# Is Skull-Vibration-Induced Nystagmus Modified with Aging?

Giampiero Neri [1],*, Letizia Neri [2], Klajdi Xhepa [1] and Andrea Mazzatenta [1]

[1] Neurosciences Imaging and Clinical Sciences Department, University of Chieti-Pescara,
66100 Chieti, CH, Italy; klajdixhepa28@gmail.com (K.X.); amazzatenta@yahoo.com (A.M.)
[2] Otolaryngology Unit, University of Insubria, 21100 Varese, VA, Italy; leti.neri@hotmail.it
* Correspondence: giampiero.neri@unich.it; Tel.: +39-3495627591

**Abstract:** Background: Despite clinical practice utilizing the Dumas test (SVINT), some questions remain unanswered, including the age-related changes in frequency (FN) and slow-phase angular velocity (SPAV). This study aims to retrospectively evaluate their variations in subjects affected by unilateral peripheral vestibular loss (UPVL). Methods: We evaluated the selected samples based on the results of the SVINT, the results of the vestibular-evoked potentials (C-VEMP and O-VEMP), and the results of the head impulse test (HIT) and we compared the results against the age of the patients. We calculated the timing between the onset of UPVL and clinical evaluation in days. The presence or absence of VEMP indicated the UPVL severity. UPVL and BPPV patients with spontaneous or pseudo-spontaneous nystagmus were compared. Results: Statistical analysis showed changes in the FN and SPAV depending on age and the side of the application of the stimulus. We also observed that, in the UPVL, the severity of the disease modifies the SPAV, but not the frequency. Conclusions: The SVINT is a simple, reliable, and straightforward test that, if evaluated instrumentally, can show significant differences with aging. Further studies need to be performed to refine the clinical significance of the test and clarify its physiological background.

**Keywords:** SVINT; unilateral vestibular disease; BPPV; aging; slow-phase angular velocity; vestibular recruitment





## 1. Introduction

The skull-vibration-induced nystagmus test (SVINT) in patients with unilateral peripheral vestibular loss (UPVL) is a quick test, which is simple to perform even in the pediatric field. It is complementary to the most commonly used tests in the clinical setting [1]. The execution technique, standardized by Dumas in 2007 [2], consists of the application of a vibratory stimulus at 100 hertz at three points of the head—precisely, on the mastoids and at the vertex or zeta point—allowing observation of the evoked nystagmus, directed generally towards the healthy ear [3]. There are, however, some exceptions, as follows: Menière disease or canal dehiscence [3–5], in which nystagmus (ny) is present in 58.9% and 100% of cases, respectively, directed towards the diseased side; some acoustic neuromas, in which the vibratory nystagmus may rarely beat towards the affected side [6–8]; finally, in central pathologies, where the Ny is most frequently down-beating [5]. In reality, the SVINT can reveal the asymmetry of the canals and otolithic labyrinth receptors [9–11]. This is possible even in cases where compensation has developed, which can reduce or cancel other signs, such as the head-shaking test (HSN) [12]. This conversely represents the alteration of the velocity storage system, induced by total or partial labyrinthine damage [13] greater than 50%. Age-related cognitive decline is also inevitably reflected in changes in vestibular tests [14–16], which do not appear to be related to the loss of function of vestibular hair cells [17] but rather seem to be related to mechanisms of central compensation and sensory substitution. Age-related variations in vestibule–ocular responses do not seem to quantitatively reflect the progressive degenerative loss of vestibular hair cells and nerves [17]. This dissociation between morphology and function, and the question of how the age-dependent

deterioration of vestibular input can compensate, are not yet clear, even if central compensation and sensory substitution are among the proposed mechanisms [18–20]. The first aim of this work was to evaluate the presence of vibratory nystagmus in elderly patients with unilateral damage. The second aim was to verify the modification of vibratory nystagmus over time in different age groups.

## 2. Materials and Methods

### 2.1. Population Studied

This is a cohort retrospective study based on records of patients observed from 2018 to 2020. A total of 67 patients were enrolled, 46 of these—engaged as the study group—referred to episodes of vertigo in the last year: 34 were affected by unilateral peripheral vestibular loss (UPVL) and 12 had been affected by benign paroxysmal peripheral vertigo (BPPV), that presented vibratory nystagmus (ny). Twenty-one subjects who had never had labyrinthine pathologies were recruited as a negative control. All participants gave informed consent before data collection; the procedure followed the current ethical laws, and we performed all tests according to the Helsinki II Declaration.

### 2.2. Exclusion Criteria

We excluded patients with psychological and neurological problems—those with prior history of vestibular pathology, migraine, kinetosis, middle ear pathology, diabetes, ototoxicity, alcoholism, and neurological problems, or without vibratory nystagmus, with vertical nystagmus, and patients who had not reported precisely the time from UPVL onset, or had not undergone all the examinations covered by this study.

### 2.3. Data Collection

All patients underwent the following:

- Collection of history, with particular attention to the timing between vertigo onset (T0) and clinical observation (T1), expressed in days;
- Skull-vibration-induced nystagmus test (SVINT);
- Head impulse test (HIT);
- Cervical and Ocular VEMPs (C- and O-VEMPs).

#### 2.3.1. SVINT

We performed the SVINT at 100 Hz with the technique described by Dumas [3] (with a commercially available system with Euroclinic Vestibular Vibrator ED500). After ocular movement calibration, the stimulus, lasting 5–10 s each, were applied perpendicularly to the skin, bilaterally over each mastoid, and centrally on frontal bone in the z point (vertex). Eye movements were recorded with video Frenzel goggles in the dark using the EDM Integrated System (Euroclinic). Our study judged the test to be positive when horizontal nystagmus, constantly beating on the same side, was evoked in at least two stimulation locations.

#### 2.3.2. Vestibular Potentials

We recorded the Vestibular potentials with a commercially available system (Labat) using air-conducted (AC) stimulation, placing a ground electrode on the sternum. Both C-VEMPs and O-VEMPs were recorded using a sampling rate of 500 Hz at 120 dB from 20 ms before stimulus onset to 80 ms after stimulus onset and averaged over 250 individual trials. Using a standard VEMPs montage, p13–n23 biphasic peaks were recorded during stimulation from ipsilateral sternocleidomastoid (SCM) muscles [21,22], while peak traces n1–p1 were recorded from electrodes positioned vertically beneath the contralateral eye. Wave absence was considered pathological in both tests.

### 2.3.3. Head Impulse Test

The HIT, recorded with ICS Impulse vHIT system Natus, was considered positive in the presence of corrective saccades, with or without gain asymmetry, and lower than 0.8°/s.

In each patient, we considered the following variables:

- The mean of slow-phase angular velocity (SPAV), calculated among all the nystagmus generated during 10 s of stimulation and expressed as "°/s".
- The mean of nystagmus frequency (FN) is calculated as the number of saccades in 10 s of stimulation and expressed in Hz (R/10—where R is the total number of saccades).

### 2.4. Statistical Analysis

UPVL and BPPV patients were analyzed for sex, age, timing and severity of symptoms to analyze the response to SVINT.

Regarding age, we split the sample into two groups: younger patients (YP—<65 yo) and older patients (OP—>65 yo).

Regarding timing, it was analyzed considering the time between the onset of vertigo and clinical observation, expressed in days (from 1 to 365 days), and the presence of spontaneous nystagmus. Regarding the spontaneous nystagmus, the sample was also divided into patients with spontaneous nystagmus where the vestibular deficit was not still compensated (early timing—ET: n = 24), and patients without spontaneous Ny where the vestibular deficit was presumably compensated (late timing—LT: n = 10). This classification observed whether the nystagmus response induced by SVINT statistically changed in the period between the onset of the deficit and the moment of clinical observation.

The severity of UPVL, expressed by the presence or absence of vestibular potentials, was calculated as being more severe in subjects without recordable VEMPs than in patients with present VEMPs. We split the 34 patients into three groups according to the disease severity expressed in the ipsilateral absence of VEMP. The group considered mild in severity had both C-VEMPs and O-VEMPs; the group believed to be of moderate severity had O-VEMPs only; the third group, considered severe, did not have either vestibular potential.

Statistical analyses were performed using commercial software Excel and open access software Jamovi; the tests performed were normality test, multivariate analysis test, and one-way ANOVA, and alpha was set as 0.05. Curve-fitting analysis was performed by using a Gaussian model, $y = y_0 + (A/(w \times \mathrm{sqrt}(pi/2)) \times \exp(2((x - x_c)/w)\hat{}2)$. Standard error (S.E.) and standard deviation (S.D.) were calculated for all analyses.

### 3. Results

In the present study, during 2018–2021, we enrolled 67 participants, comprising 46 (study group) with peripheral vertigo in the last year (34 affected by UPVL; 12 affected by BPPV) and 21 (negative control) who had never had a labyrinthine deficit.

### 3.1. Unilateral Peripheral Vestibular Loss Patients

In the UPVL group, we evaluated 34 patients, 12 females and 22 males, aged between 36 and 76 years old (average age: 61 yo), comprising 16 patients with a left vestibular deficit and 18 with a right vestibular deficit. Of these, 18 patients without spontaneous nystagmus (53%) and nine patients with spontaneous nystagmus (26%) were evaluated during the acute phase of UPVL; additionally, seven patients (21%) had suffered vestibular damage no more than two months prior, and 2 of these patients (uncompensated) had spontaneous nystagmus. HIT was pathological in all cases, with saccadic induced in the movement toward the healthy side. We divided these patients into 2 groups, 11 patients with spontaneous nystagmus (uncompensated or early time—ET) and 23 patients without spontaneous nystagmus (compensated or late time—LT). The SVINT was positive in all patients with frequency on Ny (FN) between 0.19 and 3 Hz (mean 1.76 Hz), and slow-phase angular velocity (SPAV) between 10.08 and 41.3°/s (mean 23.29°/s).

In the BPPV control group, we evaluated 12 patients, four females and 8 males aged between 39 and 75 years (average age: 63 yo). Only those patients with BPPV who presented

with positive SVINT were enrolled, regardless of the side of the pathology (6 on the right and six on the left) or the affected canal (4 left posterior, five right posterior, and 3 right lateral). The SVIN in BPPV patients had an FN between 1 and 14 Hz (average, 7.6 Hz) and a SPAV between 1.34 and 97°/s (average 24.36°/s). No patients had C-VEMP or O-VEMP absences.

In the negative control group—comprising 12 patients (9 female and 12 male), aged between 43 and 72 years old (average age: 59 yo; S.D. = 7.3), no subjects had a history of vestibular problems. Eighteen of these subjects did not have vibrational nystagmus. Only three patients (14.2%) had slight nystagmus with an FN $\leq 0.6$ n/s and a SPAV 2.6°/s (S.D. 1.4); these results were comparable to those of the literature [23]. Vestibular potentials were present bilaterally, and HIT was normal on each canal.

### 3.2. Sex

Sex was found to be a non-significant parameter in evaluating FN and SPAV, in both the UPVL and BPPV groups, which did not influence timing, severity, or age parameters.

### 3.3. Age

Considering age, the YP with UPVL was 22 (mean = $54.5 \pm 9.34$ S.D.) and the OP were 12 (mean = $72.75 \pm 2.45$ S.D.), while the YP with BPPV was 7 (mean = $55.86 \pm 8.34$ S.D.), and the OP was 5 (mean = $73.4 \pm 1.95$ S.D.). The FN and SPAV showed significant differences: the FN measures were different when applying the stimulus at the vertex or mastoid areas. In the first case, the frequency was non-significant ($p = 0.16$); in the second case, conversely, FN increased with age in both mastoids (on the left ($p < 0.05$—$F_{(1,32)} = 5.14$, mean YP $6.05 \pm 0.36$ S.E. and mean OP $7.83 \pm 0.85$ S.E.)), and on the right, ($p < 0.05$—$F_{(1,32)} = 6.3$, mean YP $7 \pm 0.53$ S.E. and mean OP $9.25 \pm 0.74$ S.E.) (Figure 1). The SPAV showed significant differences only in the spontaneous nystagmus and was higher in older patients than in younger patients ($p < 0.001$; $F_{(1,46)} = 13.08$, mean YP $3.64 \pm 0.13$ S.E.; mean OP $5.52 \pm 0.66$ S.E.). No differences were found with stimulus applied on the vertex ($p = 0.33$) or the left or right mastoids (respectively, $p = 0.96$ and $p = 0.21$) (Figure 2).

Considering the timing, the FN and SPAV showed significant differences in UPVL patients: the FN was significantly lower in LT patients than in ET patients ($p < 0.05$ ($F_{(1,32)} = 6.11$, mean ET $6.63 \pm 0.49$ S.E. and mean LT $4.3 \pm 0.83$ S.E.) when the stimulus was applied on the vertex, while no significant differences were found at either mastoids (left $p = 0.21$ and right $p = 0.89$) (Figure 3). SPAV analysis showed significant differences in both mastoids, on the left ($F_{(1,144)} = 7.13$, mean ET $22.44 \pm 1.91$ S.E. and mean LT $30.54 \pm 1.76$ S.E.) and on the right ($F_{(1,144)} = 13.99$, mean ET $17.99 \pm 1.89$ S.E. and mean LT $30.88 \pm 1.97$ S.E.), while not in the vertex ($p = 0.19$) (Figure 4). SPAV analysis only for spontaneous nystagmus showed significant differences that appeared significantly greater ($p < 0.001$) in LT than in ET ($F_{(1,46)} = 19.11$, mean ET $3.74 \pm 0.12$ S.E. and mean LP $6.2 \pm 0.98$ S.E.). Conversely, no differences were found in vertex stimulation ($p = 0.16$)—left ($p = 0.42$) and right ($p = 0.53$) (Figure 5).

The comparison by one-way ANOVA between YP and OP with UPVL and BPPV found that SPAV was highly significant only for the YP. It is greater in BPPV (mean $25.61 \pm 1.71$ S.E.) than in UPVL (mean $18.62 \pm 1.07$ S.E.) ($p < 0.001$). In the OP, the statistical analysis returned $p = 0.32$ (Figure 6).

Regarding the severity of UPVL, of the 34 patients, 6 (18%) were considered to be mild in severity, 21 (61%) were considered to be medium in severity, and 7 (21%) were considered as severe. The O-VEMP was more significant in the UPVL than C-VEMP (28 vs. 9, respectively; $p < 0.001$). Comparing VAFL and FN with severity, we found that VAFL significantly increased in SVINT (Manova, $p = <0.001$; $F_{(2,163)} = 17.5$; mild, $14.73 \pm 0.95$ S.E; moderate, $24.1 \pm 1.14$ S.E.; severe, $32.82 \pm 3.01$ S.E.), related to the severity of the damage, while the frequency does not follow this trend. Post hoc ANOVA showed $p =< 0.001$ (mild vs. moderate $F_{(1,131)} = 18.5$; mild vs. severe $F_{(1,61)} = 31.1$; moderate vs. severe $F_{(1,133)} = 10.9$) (Figure 7). In no case was the C-VEMPs absent alone. HIT was pathologic in all UPVL cases.

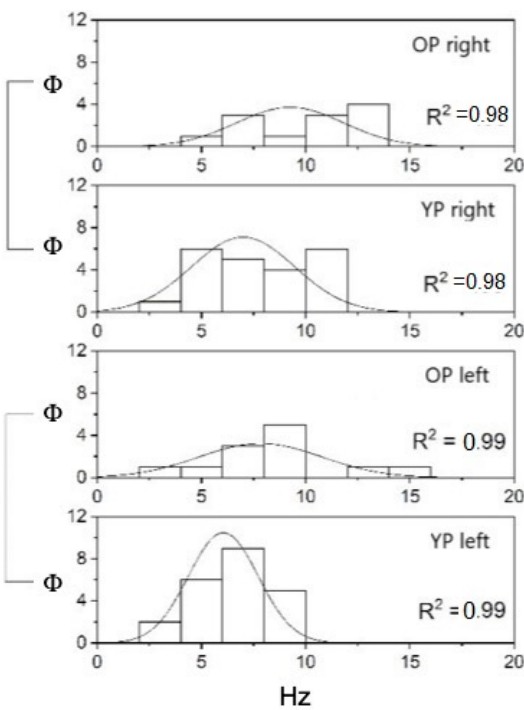

**Figure 1.** FN (Hz) in older and younger patients with SVINT applied on the right and left sides (Φ = data distribution frequency). The Gaussian peak represents the mean of SVIN frequency in Hz. Normal curve vibrational FN distribution and fitting analysis, $R^2$ = 0.99 for both left YP and OP while $R^2$ = 0.98 for right YP and OP ($p < 0.001$).

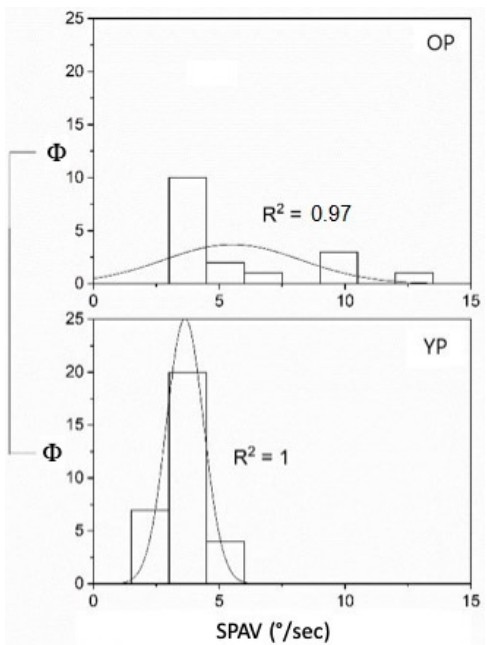

**Figure 2.** SPAV (°/s) in the spontaneous Ny in older and younger patients with UPVL (Φ = data distribution frequency). The Gaussian peak represents the mean of SPAV in °/s. fit $R^2$ = 1 for YP and $R^2$ = 0.97 for OP. Normal curve frequency distribution and fitting analysis $R^2$ = 1 for YP and $R^2$ = 0.97 OP is in ($p = 0.001$).

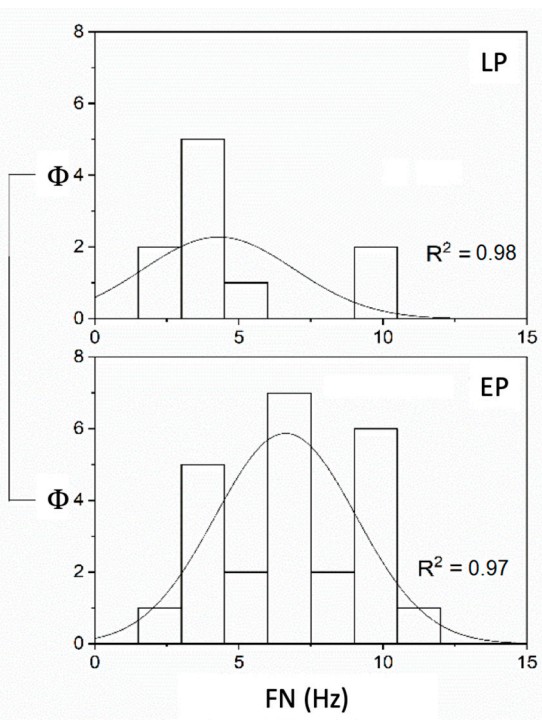

**Figure 3.** FN (Hz) in compensated (LT) and uncompensated (ET) UPVL patients with SVINT applied on the vertex (Φ = data distribution frequency). The Gaussian peak represents the mean of SVIN frequency in Hz. Normal frequency distribution curve and fitting analysis, LT, $R^2$ = 0.98, while $R^2$ = 0.97 for ET ($p$ = 0.001).

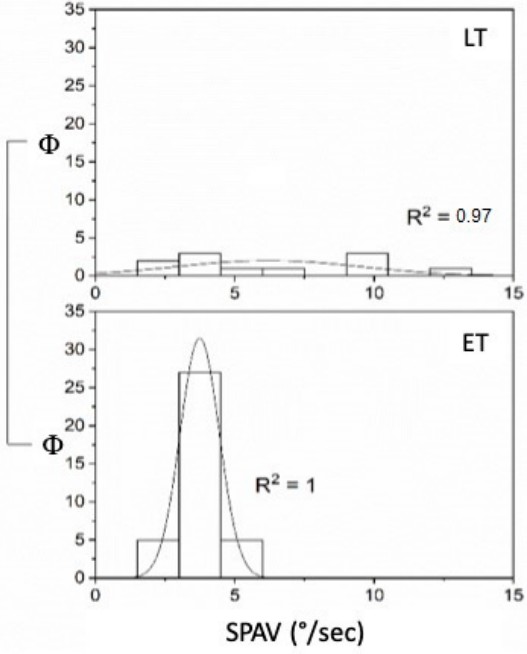

**Figure 4.** SPAV (°/s) of spontaneous Ny in compensated (LT) and uncompensated (ET) UPVL patients (Φ = data distribution frequency). The Gaussian peak represents the mean of SPAV frequency in °/s. Normal curve fit $R^2$ = 0.99 for all ($p$ = 0.001).

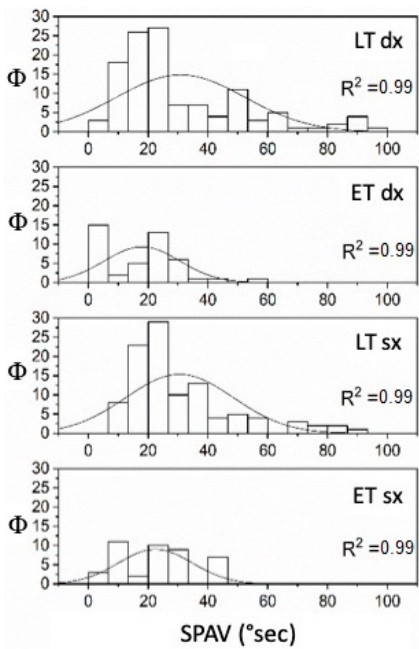

**Figure 5.** SPAV (°/s) of spontaneous Ny in compensated (LT) and uncompensated (ET) UPVL patients (Φ = data distribution frequency). The Gaussian peak represents the mean of SPAV frequency in °/s. Frequency distribution in spontaneous SPAV and normal curve fit $R^2$ = 1 for early time and $R^2$ = 0.97 for late time. Fit $R^2$ = 1 for ET and $R^2$ = 0.97 for LT ($p$ = 0.001).

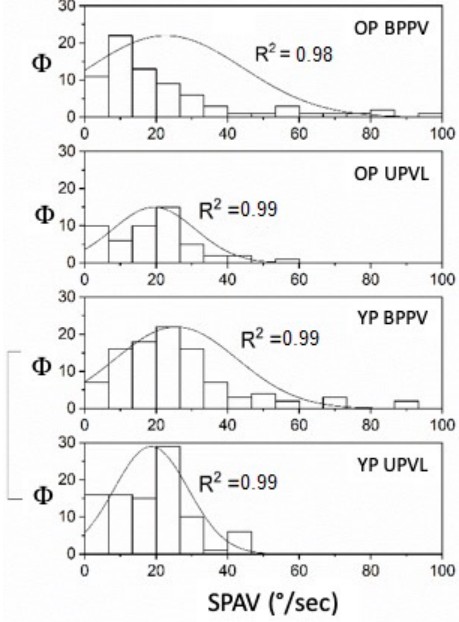

**Figure 6.** SPAV (°/s) of SVINT comparisons between UPVL and BPPV in older and younger patients (Φ = data distribution frequency). The Gaussian peak represents the mean of SPAV in Hz. Frequency distribution and fitting analysis in fit $R^2$ = 0.99 for all curves except OP BPPV, that is $R^2$ = 0.98. ($p$ = 0.001).

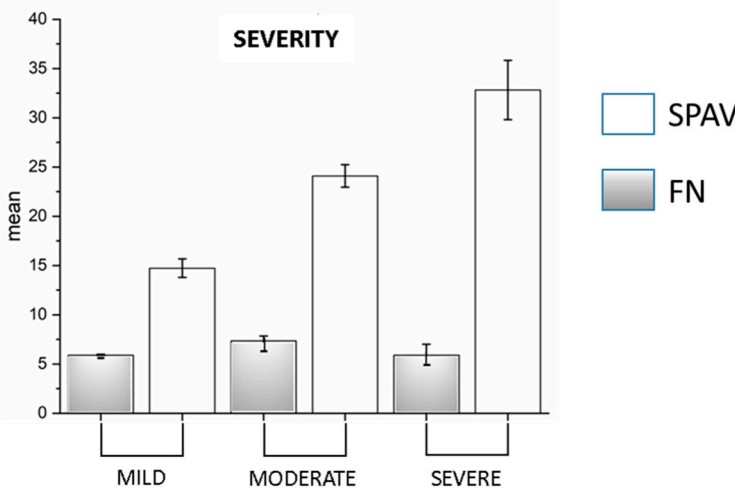

**Figure 7.** Superimposed graph of FN and SPAV in the three severity classes: mild, moderate, and severe. On the *Y*-axis are the averages of both measurements (*p* = 0.001).

## 4. Discussion

We used the SVIN test in 67 human subjects (42 males and 25 females), aged 36–76 years, to understand whether fundamental parameters can change with the patient's age. This investigation shows that SVINT was positive in all UPVL patients with FN between 0.19 and 3 Hz (average, 1.76 Hz), and all SPAV between 10.08 and 41.3°/s (average, 23.29°/s). In BPPV patients, with FN between 1 and 14 Hz (average, 7.6 Hz) and SPAV between 1.34 and 97°/s (average 24.36°/s), the results confirmed the great significance of the Dumas test. The finding that the frequency of the response to SVINT by BPPV patients was greater than the response of SPAV could be due to the absence of real vestibular damage. All the receptors in the BPPV were functional, so the result of stimulation is a result of the algebraic sum of the canal and macular excitations that are absent in the UPVL.

In elderly patients with UPVL, the frequency of SVIN increased with age only with the mastoid stimulus, while the SPAV increased only in the patients with spontaneous Ny. The SVINT applied to the mastoids in patients > 65 years of age (OP) affected by UPVL produced a response with increased frequency and reduced velocity, while not producing significant effects with the stimulus was applied to the vertex. On the contrary, considering the time between the damaging event and the instrumental evaluation (timing), the application of the vibration to the vertex reduced the nystagmus frequency. At the same time, the stimulus transmitted through the mastoid was not statistically significant. Therefore, the FN appears to be a parameter of the SVINT that can be modified over time, regardless of the subject's age, which instead interferes with the SPAV, showing a less evident but more frequent Ny in the elderly. This observation indicates that SVINT does not follow the rules of other instrumental tests. Age-related changes are present in healthy subjects and are characterized by a reduced duration of cold caloric response and of rotational VOR gain (Karlsen in 1981 [24] and Baloh in 1990 [25]), a slower saturation rate of the kinetic optical reflex, and reduced visual interference. These data in healthy patients indicate a correspondence between age-related anatomical damage and vestibular function. In addition, an alteration of postural balance has also been demonstrated in the healthy elderly, especially in situations of visual and somatosensory conflict [14,26,27].

On the other hand, this correlation with age is less evident in asymptomatic patients with UPVL, with active central compensation by vestibular damage. This result is similto those in Peterca's studies, which highlighted how the age-related trends in VOR are not consistent with the anatomic changes in the periphery, suggesting that this poor correlation between physiological and anatomical data can be due to adaptive mechanisms in the central nervous system, which are essential in maintaining the VOR [14].

Vibratory Ny is present even years after the harmful event [28]. This study demonstrates that, in elderly patients with UPVL, the frequency of Ny increases with timing only with the vertex stimulus. In our study, the frequency of the vibratory stimulation changed if applied to the mastoids or at the vertex. The first case changed with age, increasing in older patients; it reduced its response without cancellation in the second case. In addition, the SVINT seemed to be a more complex test than previously thought in this particular result. It is known that mastoid stimulation is more efficient than stimulation at the vertex for the vibrational energy transfer to the labyrinthine receptors in both sides [22]. This increase in frequency can be correlated with the central compensatory phenomena or may result from the disinhibition of cortical networks described by van Boxtel in 2006 [29]. This would be physiologically understandable in frail elderly patients in maintaining fundamental reflexes, such as the oculomotor vestibular reflex. Another possible explanation for this phenomenon could be found in the fact that the vestibular cells activated by vibration have a phase block of the action potential, similar to that found in auditory afferents [30]; therefore, the increased frequency of the Ny could be considered a sign of vestibular recruitment.

The vibrational SPAV increased in elderly patients with UPVL and younger patients with BPPV. In BPPV, vibration nystagmus is a very rare sign [31], and it is positive if BPPV is associated with an ipsilateral caloric hypofunction (Lindsay–Hemenway syndrome) [2]. Our study recruited subjects with BPPV who presented a pseudo-spontaneous Ny as a positive control and diseased patients with confirmed UPVD. The vestibular receptor was undoubtedly not damaged in the first group of patients because the treatment with liberatory maneuvers subsequently resolved the Ny. However, the presence of the Ny was an evident sign of canal dysfunction and caused by the presence of a sizeable otoconial cluster, which—through a mechanism of canal jam—led to pseudo-spontaneous Ny. This situation is similar to a canal hypofunction observed in the caloric tests [2]. Additionally, in our experience, this otolithic stimulation in BPPV patients with one horizontal semicircular canal occluded may be able to modulate SVIN, as in [32]; similarly, for UPVD, beating away from the side of the blocked canal [33] increased SPAV to spontaneous Ny in younger patients, but not in older.

SPAV is higher in compensated patients than uncompensated patients only in mastoid SVINT. The enhancement of the SPVN (by 1.5–2 times) and SVINT-related outcomes in the case of spontaneous nystagmus [34]. In our experience, this increase in SPAV is present only for the stimulus applied to the mastoid and not to the vertex and could be linked to the size of the sample.

Regarding the severity, the literature surrounding UPVL air-conducted stimuli show few abnormality rates of C-VEMPs, due to the conservation of the inferior nerve function but frequent abnormalities of O-VEMPs; in our study, also we had few patients without C-VEMPs, and a more significant absence of O-VEMPs, comparable to that already known in the literature [35]. Ohky, in 2003 [11]—in a study including 24 patients with a UPVL— found no correlation in 18 patients with an absence of C-VEMPs. This author conversely demonstrated the degree of dysfunction in the saccule is significantly related to the induction of vibratory nystagmus. Our study also observed that the SPAV increased with increasing damage, while the frequency appeared substantially the same in the various severity classes (Figure 7). This result, associated with the higher frequency of the SVIN in BPPV compared with UPVL, confirms that the functionally most essential data in vestibular damage is the SPAV on the one hand, and, on the other hand, that the frequency of SVIN is not influenced by the conservation of macular structures but of canal structures [36].

The SVIN did not change over time for the two age groups (YP–OP), which did not show significant differences. Several authors have already demonstrated that SVINT was not influenced by vestibular compensation [11,36]. Additionally, in our sample, we observed the persistence of SVIN even at a distance from the harmful event. However, while the frequency does not show significant differences with age, the SPAV of the Ny obtained in the vertex stimulation was significantly different ($p = 0.002$) between the YP group, where it was 20.17°/s, and the OP group, where it was 7.91°/s. In contrast, lateral

stimulation did not show significant differences. The reasons for this latter result have been discussed above.

## 5. Conclusions

Our investigation shows that the Dumas test is undoubtedly a simple, reliable, and useful test for evaluating vestibular imbalance. These results indicated that vibratory nystagmus in UPVL patients maintains its clinical usefulness, regardless of the patient's age, even though the fundamental parameters of the vibratory Ny—such as frequency and angular velocity—vary between the elderly and the young. The FN paradoxically increased in the older patients, mainly when the vibratory stimulus was applied to the vertex and the SPAV, especially in patients who had already obtained vestibular compensation after UPVL. Finally, the association between utricular and saccular damage leads to changes in the SVIN only for SPAV.

Despite the statistical significance of the sample, the small sample size suggests that further studies need to be performed to clarify the clinical relevance of the data and the physiological background underlying our results.

**Author Contributions:** Conceptualization, G.N.; methodology, G.N. and A.M.; formal analysis, A.M.; writing—original draft preparation, G.N. and K.X.; writing—review and editing, G.N. and L.N.; supervision G.N.; data collecting, K.X. and L.N. All authors have read and agreed to the published version of the manuscript.

**Funding:** This research received no external funding.

**Institutional Review Board Statement:** This study not require ethical approval.

**Informed Consent Statement:** All participants gave informed consent before data collection, according to the current ethical laws, and all tests were performed according to the Helsinki II Declaration.

**Data Availability Statement:** University of Chieti-Pescara, ENT Unit.

**Conflicts of Interest:** The authors declare no conflict of interest.

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
