# Peer review of "Is Skull-Vibration-Induced Nystagmus Modified with Aging?"

_audiolres, doi:10.3390/audiolres12020016_

Round 1

Reviewer 1 Report

The study provides good evidence for some not previously analyzed aspects of the Vibration test. I found some differences in the elderly population very interesting and partially non-expected, and therefore worthy of being published. I suggest a revision in the use of the abbreviations, not always clear to be understood.

I would add in the analysis of the Literature also VIII cranial nerve schwannoma in the cases of Vibration Induced nystagmus beating towards the affected side (see Modugno G, Brandolini C, Piras G, Raimondi M, Ferri G. Bone Vibration-Induced Nystagmus (VIN) Is Useful in Diagnosing Vestibular Schwannoma(VS). Sixth International Conference on Acoustic Neuroma; 2011. Los Angeles: International Conference on Acoustic Neuroma (2011).

 Dumas G, Karkas A, Schmerber S. Skull high-frequency vibration-induced
nystagmus test for vestibular function assessment in vestibular schwannoma.
In: Neuroma I, editor. Sixth International Conference on Acoustic Neuroma;
2011. Los Angeles (2011).

Hyperventilation-Induced Nystagmus in Patients With Vestibular Schwannoma. Califano, Luigi; Iorio, Giuseppina; Salafia, Francesca; Mazzone, Salvatore; Califano, Maria. Otology & Neurotology: February 2015 - Volume 36 - Issue 2 - p 303-306. DOI: 10.1097/MAO.0000000000000699).

Author Response

I thank the reviewer for his valuable suggestions, which I have very gladly applied in the corrections. In particular, I added the note regarding the vibratory test in neuroma and reduced the abbreviations as much as possible. The manuscript underwent also  extensive English revisions checked and corrected by a native English-speaking colleague.

Reviewer 2 Report

Comments about « Is SVINT modified with ageing?” from G.Neri et al.

Minor typing errors

Abstract line 10 “Slow Phase Angular velocity (SFAV)” should be “…(SPAV)”

Text between Lines 25 to 31 must be deleted

Introduction  line 36 “dumas” should be “Dumas”

Line 39 : delete the second “towards”

Line 56: Ny was not defined

Line 63 : BPPV was not defined

Line 60 :”Sixtyseven” should be: :”Sixty seven”

Material & methods line 71 “UPVC” should be “UPVL”

Line 74 “all patients was” should possibly be “all patients were”

Line 98 “”..correctives saccades” should be “corrective saccades”

Results Line 128 “”…where affected..” should probably be “..were affected..”

Line 135”..and SPAV 2.6°/s (SD 1.4)” could possibly be” and SPAV < 2.6°/s (SD 1.4)..”

Line 139”in the UPVL group.. 12 female and 22 males..” should be” .. 12 females and 22 males..”

Line 151 : why n/s and not Hz ?

Line 166 167 177 : what is F1,32 ?

Line 172 182 : please define F1,46

Line 167 “Fig 1 show” should be: “Fig 1 shows” idem line 179

Line 173 : “p=93” should be “.93” idem for right

Discussion “ The SVINT are tested in..” might be “The SVINT has been tested in ..”

Line 305 “Regarding the others parameters..” should be: “Regarding the other parameters”

Line 308 :”..that is most high in ..” might possibly be “ .. that is much higher in..”

Line 309: “these differences regards the ..” should be “these differences regard the ..”

Line 317 “OKI” should be “Ohki”

Line 326 “VAFL” is not explicated or mis-orthographied

Remarks   Abstract line 15 “ ..BPPV with spontaneous ny..” is not clear and could be explained or rephrased ( does it consider the Hor BPPV with a pseudo Spont Nyst which disappears when the head is slightly tilted downward?) or a Lindsay Hemenway syndrome...

The sentence line 16 – 18 is not clear and could be rephrased

Line 19 conclusion “ the SVINT is a simple… can show significant differences that still…” do you want to tell: “..significant differences with ageing that still..”? can you notify it more precisely?

Introduction line 55 “greater than 50%” of what ?( caloric test hypofunction or else?)

Methods and results : I don’t understand why you compare left and right ear : it is more exact to compare pathological vs normal ear. For comparison of mastoids and with vertex, a statistical paired test adapted must be used.

Results:  the mean age of the control population is not notified. (line 133-135)

A characteristic of population table by groups OP YP can be interesting, it is important to show to the reader the potential variable as value of deficit (HIT and cVemp) which can biased your results between the groups. In your multivariate analysis, you don’t precise what are the variables used (statistically significant or not) and eventually their link (VEMP could be correlated to HIT so eventually redundancy) in your model.

Units of measure have to be mentioned in the different results (line 166-196)

The sentence line 136 – 138 must be clarified and possibly be cut in 2 parts. One part concerning normal control patients and the other one the BPPV patients

Lines 139 to 140  the age and numbering of the 34 patients with UVL have already be mentioned  line 128 and is redundant ( to be deleted in at least one of these 2 places)

In your results the SPAV of SVIN is much impressing in your population of BPPV (97°/s) How do you explain it? Could it be a Lindsay Hemenway syndrome? There is partially explained in the discussion

For more clarity could you notify  on line 153 to what exact population are addressed these results about SVIN?”in all patients “ is it all patients with UVL or all patients UVL +patients  BPPV?

The sentence Line 163 to 164 is not clear and might be rephrased.

Could you explicite line 176 the terms  EP and LP (they are used for the first time and have not been detailed before)

Line  179 after “right (p=.89)” the word “mastoids” might be added

Figures : for more clarity and more easy use of graphs the parameters and categories analyzed should be noticed on the different axis (X and Y axis)(number of patients , FN in n/s or Hz or SPAV in °/s…)

Fig 4 and Fig 5 are difficult to understand: Fig 4  the legend mention a comparison between YP and OP and on the fig it is mentioned LP and EP…

The English language might  be in some places improved

These results are globally interesting and it is an original work . Some data  mention that in older patients the SVIN freq is higher but that the SPAV is lower in long lasting patients . Conversely you mention line 170 – 175 that at the acute period of the UVL (“the spontaneous nystagmus one’s”)older patients have higher SPV ( possible attenuation with ageing of cerebellar control and adaptation)

One other  interesting point is that You did not observe relationship between SVIN FN or SPAV and VEMPs absence . this could be discussed in the light of what is currently admitted and have already been discussed : the Nystagmus observed with SVIN is more relevant from Canal contribution than from otolith contribution ( Dumas et al “How to do & why perform the SVIN” European annals 2021.

However the discussion is possibly too long and should be possibly clarified

Author Response

I thank the reviewer for his valuable suggestions, which I have very gladly applied in the corrections. The manuscript underwent extensive English revisions regarding minor typing errors checked and corrected by a native English-speaking colleague.

1) I agree to use pseudospontaneous ny because it clarifies the sentence's meaning.
2) We have clarified lines 16-18
3) Line 19: we have added the reference to age
4) Line 55 we have modified with "greater than 50%."
5) Method. We did not compare the right and left sides but the results obtained from the stimulation of the three skull points separately about the parameters age, severity, and timing.
6) line 133-135, we added the mean age of the control population
7) we have joined the measurement unit in the text and figures. We have already canceled the redundant sentences and substantially modified all text to comprehend better. The relationship between SVINT and VEMPs absence, in reality, is valid for FN but not for SPAV, and I have joint fig 7 for clarifying this result.

finally, we have joined the bibliography you have suggested that have better improved the paper

Reviewer 3 Report

This is a very interesting and needed work to address the issue of age and SVIN.

The authors have gathered several patients to analyze and found some interesting correlations.

I am ion genial surprised to see the SPAV of SVIN is higher in patients with BPPv then in patients with UPL. Is this due to the characteristics of the patients?

Some minor commets

Material and methods. Section 2.3.3 should be in statistical analysis

The interesting issue begins at line 156. All before written results could go in a table with demographic data.

Do not use paper-work initials ("ny") and use always the same ones (SPAV, appears as several times different) .

Why sex is not significant?? test performed not mentioned neither p.

Severity. Same as before: you have to mention statistical test that gives the considerations.

Age. All data in demographic table (I do not see age of normals). Line 164: "frequency is non significant" do you mean "age"? Line 170: SPAV? do you mean SPVN?. 

Discussion....extremely verbose...misses the target of the paper and question and is hard to follow at times. I suggest to put into order the interest and purpose of the paper as stated at thee introduction.

Author Response

Thanks for your gratifying review. We agree with all your criticisms and considerations. We believe that the difference in SPAV of SVINT could be due to receptor damage, present in UPVL and absent in BPPV patients. This consideration was present in the text.
We hame uniformed the initials and substantially modified the text. Native English speakers revised the English. We have eliminated some parts in the discussion to reduce the wordy sentences.
Statistically, all the parameters confronted to sex do not shown significant results.

Round 2

Reviewer 3 Report

Th authors have done a good work

Author Response

Dear Reviewer,
I very much appreciated the attention with which you have read the work and the corrections you have suggested to improve it and with which I fully agree. I edited the text according to your suggestions. In particular, I modified the conclusions in two points by adding the final sentence "Despite the statistical significance of the sample, the reduction in sample size suggests that further studies need to be performed to clarify the data's clinical relevance and the physiological background underlying our results."
I want to thank you for the work you have done. Given the substantial contribution you have made to this writing, I would be honored if you would like to add your name to the list of authors.
Sincerely
Giampiero Neri